# Deciphering Common Traits of Breast and Ovarian Cancer Stem Cells and Possible Therapeutic Approaches

**DOI:** 10.3390/ijms241310683

**Published:** 2023-06-26

**Authors:** Ivan Lučić, Matea Kurtović, Monika Mlinarić, Nikolina Piteša, Ana Čipak Gašparović, Maja Sabol, Lidija Milković

**Affiliations:** 1Laboratory for Oxidative Stress, Division of Molecular Medicine, Ruđer Bošković Institute, 10000 Zagreb, Croatia; ivan.lucic@irb.hr (I.L.); monika.mlinaric@irb.hr (M.M.); ana.cipak.gasparovic@irb.hr (A.Č.G.); 2Laboratory for Hereditary Cancer, Division of Molecular Medicine, Ruđer Bošković Institute, 10000 Zagreb, Croatia; matea.kurtovic@irb.hr (M.K.); nikolina.pitesa@irb.hr (N.P.); maja.sabol@irb.hr (M.S.)

**Keywords:** breast cancer, ovarian cancer, cancer stem cells (CSC), CSC markers, signaling pathways, tumor microenvironment, cancer immunoediting, in vitro and in vivo models, CSC-targeted therapy

## Abstract

Breast cancer (BC) and ovarian cancer (OC) are among the most common and deadly cancers affecting women worldwide. Both are complex diseases with marked heterogeneity. Despite the induction of screening programs that increase the frequency of earlier diagnosis of BC, at a stage when the cancer is more likely to respond to therapy, which does not exist for OC, more than 50% of both cancers are diagnosed at an advanced stage. Initial therapy can put the cancer into remission. However, recurrences occur frequently in both BC and OC, which are highly cancer-subtype dependent. Therapy resistance is mainly attributed to a rare subpopulation of cells, named cancer stem cells (CSC) or tumor-initiating cells, as they are capable of self-renewal, tumor initiation, and regrowth of tumor bulk. In this review, we will discuss the distinctive markers and signaling pathways that characterize CSC, their interactions with the tumor microenvironment, and the strategies they employ to evade immune surveillance. Our focus will be on identifying the common features of breast cancer stem cells (BCSC) and ovarian cancer stem cells (OCSC) and suggesting potential therapeutic approaches.

## 1. Introduction

Breast cancer (BC) and ovarian cancer (OC) are the most common and sixth most common cancers, respectively, and the first and fourth leading causes, respectively, of cancer-related deaths among women under the age of 60 [1,2]. Both are also the deadliest women’s cancers, BC altogether and OC among gynecologic malignancies.

Interestingly, although BC and OC are two different cancers that arise in different tissues of the body, women with breast cancer susceptibility gene type 1 and type 2 (*BRCA1* and *BRCA2*) mutations have a higher risk of developing BC and/or OC. The specific factors that determine whether a woman with a *BRCA1/2* mutation will develop breast or ovarian cancer are not fully understood. Since *BRCA1* and *BRCA2* are involved in DNA repair, mutations in these genes can cause genetic errors leading to cancer. Yet, the specific effects of *BRCA1/2* mutations on cancer development can vary depending on the type of mutation, its location within the gene, and other genetic and environmental factors [3,4]. Cancers related to *BRCA1/2* mutations comprise about 5–7% of BC and 20–25% of OC cases [5]. The cumulative risk of developing BC or OC by age 80 is 72% for *BRCA1* and 69% for *BRCA2* mutation carriers in BC, whereas for OC it is 44% for *BRCA1* and 17% for *BRCA2* mutation carriers [6] and estimated to be even higher [7]. While these women may never develop cancer, as women without a *BRCA1/2* mutation may also develop BC or OC, risk-reducing surgeries such as bilateral mastectomy, bilateral salpingo-oophorectomy, or a combination of both procedures are suggested to reduce the risk of developing BC or OC. They have been shown to reduce the risk of cancer that arises in the removed tissue by approximately 90%. While salpingo-oophorectomy has also been shown to reduce the risk of BC [8,9,10,11,12], the data should be evaluated cautiously [13]. Indeed, women over 45 with *BRCA1/2* mutations have a three-fold increased risk of BC if taking hormone replacement therapy after salpingo-oophorectomy [14]. Therefore, the decision for women with a *BRCA1/2* mutation to undergo these surgeries should be based on individual factors such as age, family history, personal preferences, and overall health, which requires a patient-oriented approach. 

The majority of BC (~99%) and OC (~90%) are of epithelial origin, arising from epithelial cells lining the lobules and terminal ducts in the breast or surrounding the ovaries. OC is histopathologically classified into five main types: high-grade serous carcinoma (HGSOC), low-grade serous carcinoma, endometrioid carcinoma, clear-cell carcinoma, and mucinous carcinoma [15]. BC is highly heterogeneous and mainly classified according to molecular features (hormone receptors, HER2, and Ki67) into luminal A, luminal B, human epidermal growth factor receptor 2 (HER2)- enriched, and basal-like, which is a subtype of triple negative BC (TNBC) [16]. Expression of hormone receptors can also be used for clinically applicable classification of HGSOC [17]. 

Cancer recurrence is a major concern for patients with BC and OC, and several factors influence its occurrence, such as age, tumor grade, tumor size, axillary nodal involvement, and hormone receptor status [18]. Recurrence is discovered in 25% of early detected OC and in 80% of more advanced OC [19], while in BC it varies between 10 and 41%, depending on subtype and other factors [20,21]. Additionally, the five-year survival rate for recurrent OC is less than 30% [22], and 24% for metastatic BC [23]. Therefore, it is crucial to gain a better understanding of the mechanisms that drive tumor survival and expansion. Cancer progression and recurrence are mainly attributed to subpopulations of cancer cells that are resistant to cancer therapy and have the ability to grow the tumor. These cells are referred to as cancer stem cells (CSC) or tumor-initiating cells. 

In this review, we will discuss the distinctive markers and signaling pathways that characterize CSC, as well as their interactions with the tumor microenvironment (TME) and the strategies they employ to evade immune surveillance. Our focus will be on identifying the common traits of breast cancer stem cells (BCSC) and ovarian cancer stem cells (OCSC), along with potential therapeutic approaches.

## 2. Cancer Development and Progression

Ovarian cancer is characterized by rapid proliferative growth and great metastatic potential. Indeed, epithelial–mesenchymal transition (EMT) is a characteristic of ovarian epithelium cells. During this process, epithelial cells depolarize, disassemble cell–cell contacts, acquire fibroblast-like morphological features, and adopt an invasive, migratory phenotype [24]. It was suggested that mutations in tumor protein P53 (*TP53*), RB transcriptional corepressor 1 (*RB1*), and phosphatase and tensin homolog *PTEN*, are necessary for initiating the transformation of ovarian surface epithelium stem cells [25].

If OC is not detected at an early stage (Stage I), before cancer spreads beyond the ovaries, the survival rate of patients decreases significantly. OC can reach stage II—metastasized to pelvic organs (the uterus, fallopian tube, bladder, and rectum), stage III—metastasized across the pelvic cavity to abdominal organs (the omentum, small intestine, and retroperitoneal lymph nodes), or stage IV—metastasized beyond the peritoneal cavity to distant parenchymal organs (liver and lung), lymph nodes, bones, and brain [15,26]. Non-specific symptoms, which usually occur when cancer has already progressed to an advanced stage, are the main reason for the high mortality rates of OC. Unfortunately, more than 70% of OC cases are diagnosed at a late stage [27]. In the case of epithelial ovarian cancer (EOC), metastasis can occur via three different routes: the transcoelomic, hematogenous, or lymphatic routes. Transcoelomic metastasis is the most common [27,28]. Transcoelomic metastasis occurs through the build-up of ascites fluid within the peritoneal cavity. In this process, metastatic OC cells undergo EMT and enter the ascites fluid by shedding from the primary tumor as single cells or in groups as spheroids and spread passively [29,30]. Once a colony is established at a secondary site, the cells can undergo a mesenchymal-to-epithelial transition (MET) and begin to grow rapidly. Research has shown that ascites is enriched in OCSC [31,32], which have the ability to self-renew, differentiate, and resist anoikis [33], thus playing a pivotal role in the formation of multicellular spheroids during the transcoelomic peritoneal dissemination [29].

Breast cancer typically originates from the epithelial cells within the ducts (85%) or lobules (15%) of the breast glandular tissue [34]. Recent studies suggest the existence of a common epithelial progenitor or stem cell located at the terminal duct lobular units (TDLUs), which exhibits a high degree of phenotypic plasticity and is responsible for the development of BC [35]. Initially, BC growth is limited to the duct or lobule and is considered “in situ”. At this stage, the cancer is typically asymptomatic and has minimal potential for metastasis. However, over time, these in situ cancers may progress and invade the surrounding breast tissue, leading to invasive BC. If left untreated, invasive BC can spread to nearby lymph nodes (regional metastasis) or other organs in the body (distant metastasis). Cancer metastasis is a complex process that involves multiple steps, including tumor cell dissociation, neoangiogenesis, intravasation, survival and diffusion through the circulation, adhesion to target tissues, extravasation, and establishment of metastatic foci. BC metastasis follows an organ-specific pattern, with a preference for bone, liver, lung, and brain. Metastatic BC is highly heterogeneous, making it challenging to identify risk factors and treatments [36]. Genomic analysis of synchronous primary BC and metastases which were not exposed to therapy revealed differences in their repertoire of somatic mutations. While the driver genes including *TP53*, phosphatidylinositol-4,5-bisphosphate 3-kinase catalytic subunit alpha (*PIK3CA*), and GATA binding protein 3 (*GATA3*) were common in both sites, EMT-related genes were enriched in or restricted to metastases. This mutational difference contributes to spatial intratumor heterogeneity [37].

BC and OC share approximately 50% of their top mutated genes. Integrated molecular analysis revealed a great difference between BC subtypes and many commonalities between HGSOC and basal-like TNBC, indicating related etiology [38]. The main commonalities include alterations in *TP53* and *BRCA1* with additional *PTEN* loss, which is an early initiating event in *BRCA1*-associated basal-like BC [39] and a known putative driver in OC [40]. In addition, the synergistic effect of two BRCA1 associated RING domain 1 (*BARD1*) mutations contributes to the tumorigenesis of *BRCA1*-associated BC and OC [41], and four new shared driver genes for BC and OC have recently been proposed [42].

## 3. Biomarkers of Breast Cancer Stem Cells (BCSC) and Ovarian Cancer Stem Cells (OCSC)

BC and OC are highly heterogenous, comprising genetically and molecularly diverse cells. Different theories explain how cancer develops and progresses. According to the stochastic theory, all cancer cells can contribute by accumulating random mutations, while the hierarchy theory suggests that CSC control proliferation and progression [43]. 

CSC possess the ability to self-renew and undergo symmetrical/asymmetrical division, which contributes to tumor heterogeneity, tumorigenicity, dissemination, and resistance to therapy [44]. While chemotherapy or radiotherapy can eliminate a large proportion of tumor cells, the presence of resistant CSC underscores the need for more extensive research to discover more effective treatments. Due to low abundance in tumor bulk (estimated 0.1–1% in BC), identifying CSC depends largely on specific markers [45]. 

CD44, CD24, CD133, aldehyde dehydrogenase 1 (ALDH1), CD326 (EpCAM), CD338, and the pluripotency markers NANOG, OCT4, and SOX2 are shared among BCSC and OCSC, although there are others (Table 1).

In a seminal research study, Al-Hajj et al. identified BCSC by showing that 100 CD44+/CD24−/low-BC cells formed tumors in immune-deficient NOD/SCID mice, while even 100 times more cells of a different phenotype failed to do so. While these BCSC are a minority, they can generate a diverse range of differentiated cells that form the bulk of the tumor [45]. **CD44** is a transmembrane glycoprotein that binds hyaluronic acid and other extracellular ligands, facilitating activation of diverse signaling pathways to regulate cell proliferation, adhesion, migration, survival, invasion, and EMT [46,47]. **CD24** is a cell-surface protein involved in cell-matrix and cell–cell communication [48], the expression of which is a prognostic marker in BC as well as in OC [49,50]. While in BC, the expression of CD24 correlates with more differentiated, epithelial characteristics and a lower expression is characteristic of basal-like BC [51], in OC, the expression of CD24 is associated with higher metastatic traits, chemoresistance, and poor prognosis [52,53]. Differential expression of CD24 and CD44 between OC and normal epithelium highlights CD24 as a specific marker for the recurrence of OC [54], whereas CD44+ cells have been shown to be resistant to standard chemotherapy used for OC treatment, such as carboplatin and paclitaxel [55]. High expression of CD44 in BC is mainly linked with metastasis and not tumor initiation [56]. Therefore, inhibition of CD24 and CD44 might be therapeutically relevant. Knockdown of *CD24* in SKOV3 cells suppressed OC growth in mice in vivo [54]. An anti-CD44 antibody approach reduced invasion and cell migration of ER+ BC cell lines and TNBC cell lines [57]. Indeed, BIWA-4 (bivatuzumab), an anti-CD44 antibody, is currently in preclinical trials for patients with head and neck tumors, indicating the interest in CD44 inhibition as a potential cancer therapy [58]. 

The **CD44+/CD24−/low** phenotype is shown to distinguish both BCSC and OCSC. Meng et al. have demonstrated that the CD44+/CD24− phenotype serves as a marker for OCSC, as these cells exhibit increased differentiation, invasion, and resistance to chemotherapy [59]. Similarly, CD44+/CD24−/low cells show increased chemoresistance [60] and radioresistance [61] in BC. In addition, the CD44+/CD24−/low phenotype may not necessarily correlate with clinical outcomes due to the tumor-type-dependent frequency of BCSC [51,62,63]. However, when a tumor has more than 10% of BCSC, it is correlated with worse clinical outcomes [62]. High CSC plasticity, the ability to transform and change in response to various signals from the TME, including hypoxia, transforming growth factor beta (TGFβ, and epidermal growth factor (EGF)) contribute to tumor growth [64]. The shift from CD44−/low to CD44+/high is possible and has been linked to a more aggressive phenotype [65,66,67]. CD44+/CD24−/low BCSC play a crucial role in cancer invasion, as cell lines with a higher proportion of CD44+/CD24−/low cells exhibit greater expression of pro-invasive genes, such as interleukin-1 alpha (*IL1α*), interleukin-6 (*IL6*), and interleukin-8 (*IL8*) [65]. Though CD44 was implied as a homing receptor [56], Sheridan et al. found that the CD44+/CD24− phenotype is not enough for homing of cancer cells at the site of metastasis and its proliferation, implying the need for other factors for successful metastasis [65]. Among them, and shared by BCSC and OCSC, are ALDH1, CD133, CD326 (EpCAM), CD338, and others listed in Table 1.

**ALDH1** is an enzyme that oxidizes aldehydes and retinol to retinoic acid (RA). The oxidation of toxic aldehydes protects cell while the activation of RA signaling generally affects cell development, differentiation, and apoptosis, and promotes cancer cell proliferation, drug resistance, and inhibition of apoptosis [68]. ALDH1 is a marker of OCSC and BCSC associated with cell proliferation, migration, poor survival, and chemoresistance [66,69,70,71,72]. 

**CD133** is a transmembrane single-chain glycoprotein also known as prominin, mainly located in cholesterol-rich membrane domains (lipid rafts) [73]. In normal breast tissue, CD133 is not recognized as a stem cell marker [74], but expression of CD133 in *BRCA1*-associated BC is correlated with higher tumorigenicity in mouse mammary tumors [75]. Perturbations in TME, namely hypoxia, increase the expression of CD133 [76]. CD133+ cells in TNBC show higher self-renewal properties as well as a capacity of transdifferentiation, thus contributing to vasculogenic mimicry [77]. CD133 is also a widely described OCSC marker associated with chemoresistance, elevated migration, and invasion ability [78,79].

**CD326**, known as the epithelial cell adhesion molecule (EpCAM), is a transmembrane glycoprotein involved in cell adhesion. The higher expression of CD326 in BC is associated with the stem-like phenotype, increased invasiveness, bone metastasis, and radioresistance [80,81]. Similarly, the overexpression of CD326, mainly observed in metastatic OC, contributes to EMT leading to metastasis [82]. Thus, CD326 is considered as a possible therapeutic target for EOC and metastatic BC. Zheng et al. showed that simultaneous targeting of CD326 and CD44 blocks ovarian intraperitoneal tumor outgrowth more efficiently than single targeting [83]. While adecatumumab (MT201), a fully human anti-EpCAM antibody, showed significant inhibition of MCF-7 breast cancer cell proliferation even without a complement and immune cells [84], in a randomized phase II trial, its monotherapy use in patients with metastatic BC, albeit showing some benefit in patients overexpressing EpCAM, did not lead to tumor regression [85]. Catumaxomab, an antibody against CD326, entered clinical trials in patients with advanced OC [86]. 

**CD338** is an ABC transporter often overexpressed in OCSC and associated with chemoresistance [87,88,89]. Overexpression of CD338 was shown to be unique for the luminal progenitor subpopulation of *BRCA1*-mutated cells [90]. Other common markers, such as receptor tyrosine kinase-like orphan receptor 1 (ROR1), LIN28, CD90, leucine-rich repeat-containing G-protein-coupled receptor 5 (LGR5), CD49f, Spalt-like transcription factor 4 (SALL4), and B-cell-specific Moloney murine leukemia virus integration site 1 (BMI1), also emerged and are shown in Table 1.

A more accurate identification of CSC relies on a combination of various markers, particularly due to their distinct distribution patterns and observed stemness and tumorigenic potential. The percentages of BCSC as well as expression of their markers, such as CD44, CD24, ALDH1, and CD133, are tumor-subtype dependent. Basal-like and HER2+ tumors exhibited a higher frequency of ALDH1+ cells compared to luminal tumors, and a higher percentage of CD44+/CD24− cells is mainly attributed to TNBC and to *BRCA1*-mutated BC [51,91]. TNBC cell lines with the ALDH1+/CD44+/CD24− and ALDH1+/CD44+/CD133+ phenotypes exhibit increased malignant characteristics, both in vitro and in vivo [92]. Similarly, Kryzcek et al. demonstrated that ALDH1+/CD133+ OCSC formed three-dimensional spheres more efficiently than single positive ALDH1+ or CD133+ cells [93].

**NANOG, SOX2, and OCT4** are well-known transcription factors essential for maintaining self-renewal and pluripotency in both normal and cancer stem cells, including BCSC and OCSC. Overexpression of NANOG promotes EMT, cell migration, and invasion in OC and is often associated with high-grade cancers, serous histological subtypes, reduced chemosensitivity, and poor overall and disease-free survival [94]. Furthermore, inhibition of NANOG has been shown to reduce proliferation and migration in BC cells [95]. SOX2 expression is associated with sphere formation, tumor initiation, cell proliferation, migration, drug resistance, and the expression of stemness-related and EMT-related genes in BC and OC [96,97,98]. OCT4 expression is linked to chemoresistance, tumorigenicity, and poor outcomes in BC and OC [99,100]. These factors are commonly co-expressed in CSC and play significant roles in promoting cancer progression, including EMT, migration, and invasion.

Aside from common markers, some divergent markers of BCSC and OCSC can be found in Table 1.

**Table 1 ijms-24-10683-t001:** Markers of BCSC and OCSC.

Common Markers	Biological Function in BCSC and OCSC	Reference
CD44	Transmembrane glycoprotein associated with chemoresistance.	[45,101]
CD24	Cell-surface glycoprotein associated with metastasis, chemoresistance, and poor prognosis.	[49,52]
ALDH1	Aldehyde dehydrogenase, an enzyme involved in oxidation of aldehydes, associated with cell proliferation, migration, poor survival, and chemoresistance.	[69,72]
CD133	Transmembrane glycoprotein associated with chemoresistance, elevated migration, and invasiveness.	[75,78]
CD326	Glycoprotein important for calcium-independent cell adhesion.	[80,83]
CD338	Adenosine triphosphate (ATP)-binding cassette transporter associated with chemoresistance.	[90,102]
NANOG	Transcription factor essential for maintaining self-renewal and pluripotency.	[94,95]
SOX2	Transcription factor important for cell proliferation, migration, drug resistance, and expression of stemness-related and EMT-related genes.	[96,98]
OCT4	Transcription factor often co-expressed with NANOG and SOX2 associated with increased drug resistance and tumorigenicity.	[31,103]
ROR1	Receptor tyrosine kinase important for proliferation, migration, and invasion.	[104,105]
LIN28	RNA-binding protein important for reprogramming of somatic cells to induced pluripotent stem cells.	[106,107]
CD90	Membrane glycosylphosphatidyl inositol (GPI)-anchored protein, related to tumor initiation and aggressiveness, and poorer patient prognosis.	[108,109]
LGR5	Transmembrane receptor, increases the stemness of BC cells, BC recurrence, poor outcome, and high tumorigenicity. It promotes EOC proliferation, metastasis, and EMT. Yet, its high expression in HGSOC is linked with improved progression-free survival.	[110,111,112]
CD49f	Increases tumorsphere-formation ability, enhances tumorigenicity, drug resistance, and self-renewal properties.	[113,114]
SALL4	Transcription factor, involved in cancer cell stemness, invasion, proliferation, tumor aggressiveness, and poor survival.	[115,116]
BMI1	Regulator of gene expression, necessary for self-renewal properties of stem cells, plays a role in tumor aggressiveness, invasion, EMT, and drug resistance.	[117,118]
**BCSC** **Markers**	**Biological Function in BCSC**	**Reference**
CD61, ESA	Increases tumorsphere-formation ability, enhances tumorigenicity, drug resistance, and self-renewal properties.	[113]
Sca-1	Enhances cancer cell tumorigenic ability.	[119]
CD70	BCSC self-renewal, metastasis, and tumorgenicity.	[120]
CD29	Integrin protein, enhances metastatic potential, EMT, and cell migration.	[121]
KLF4	Transcription factor, maintenance of breast cancer stem cells, cell migration, and invasion.	[122]
**OCSC** **Markers**	**Biological Function in OCSC**	**Reference**
CD117	Receptor tyrosine kinase, promotes self-renewal, differentiation, and regeneration of tumor in xenograft model, chemoresistance, and metastasis.	[123]
CD166	Glycoprotein, increases sphere-forming ability, adhesion, cell migration, and chemoresistance, high tumorigenic potential.	[124]
CD184	Chemokine receptor, important for migration and proliferation.	[125]
CD243	ABC transporter, responsible for paclitaxel resistance in OCSC.	[126]
ETRA	Endothelin receptor-A, ETRA inhibition prevents chemotherapy-induced increase in CSCs, reduces the formation of tumor spheres.	[127]
NPRA	Atrial natriuretic peptide receptor, associated with CSCs induced tumorigenesis.	[128]
ZIP4	Transmembrane zinc transporter, marker for tumor formation in vivo, self-renewal, and differentiation abilities in vitro.	[129]
IL-17R	Promotes self-renewal of CD133+ CSC in OC by binding IL-17 produced by the tumor microenvironment.	[130]
MISRs	Müllerian-inhibiting substance receptors, overexpressed in CD44+CD24+EpCAM+ cells of various OC cell lines.	[131]
c-MYC	Transcription factor, involved in reprogramming of OCSC through interaction with the tumor microenvironment.	[31]
MyD88	Adaptor protein associated with Toll-like receptor (TLR) signaling, resistance to proapoptotic signals, and the ability to create a pro-inflammatory microenvironment.	[132]
SNORA72	Highly expressed in ovarian sphere cells with CSC-like characteristics.	[133]

## 4. Signaling Pathways in BCSC and OCSC

A great number of oncogenic signaling pathways has been shown to be implicated in the maintenance of BCSC and OCSC (Figure 1); therefore, targeting these pathways represents a promising therapeutic approach to combating the CSC population.

**WNT signaling** is an evolutionary conserved signaling pathway that plays a crucial role in normal embryonic development and adult homeostasis of various organs, including the mammary gland and ovary [134,135]. WNT signaling involves the binding of the WNT ligand to LRP5/6 and Frizzled receptor, which inhibits β-catenin phosphorylation by glycogen synthase kinase 3 (GSK-3) and subsequent proteasomal degradation. The accumulated β-catenin translocates from the cytoplasm to the nucleus, where it regulates cyclin D1 expression [136]. Aberrant regulation of this pathway due to mutations and deletions of genes is common in cancer. WNT promotes EMT and chemoresistance in OCSC through induction of SNAIL and ATP-binding cassette super-family G member 2 (ABCG2) transporter pump expression, respectively [137,138]. Additionally, WNT signaling, activated by WNT5b ligand secretion from macrophages, increases the ALDH+ CSC and immunosuppressive traits of macrophages [139]. The pathway is overactive in 60% of BC [140] and can affect self-renewal, differentiation, metastasis [141], and therapy resistance [137,142]. In HER2+ BC, HER2 interacts with a serine/threonine protein kinase B (AKT) and extracellular signal-regulated kinase (ERK) to inhibit β-catenin, leading to trastuzumab resistance and EMT promotion [142]. Suppression of the pathway reduces BCSC properties, migration, and growth in TNBC [143]. Silencing β-catenin reduces ALDH1+ BCSC and increases sensitivity to chemotherapeutics [144,145]. Hypoxia-inducible factor (HIF)-2α overexpression, induced by chronic hypoxia, supports the BCSC phenotype and induces paclitaxel resistance through activation of the WNT/β-catenin pathway, which can be reversed by DKK, a WNT pathway inhibitor [146].

**The PI3K/AKT/mTOR pathway** plays a crucial role in regulating cell proliferation, metabolism, angiogenesis, and differentiation. Phosphatidylinositol-3-kinase (PI3K) is activated by the binding of growth factors and cytokines to the receptors on a cellular membrane, such as epidermal growth factor receptor (EGFR), insulin-like growth factor I receptor (IGF-IR), and fibroblast growth factor receptor (FGFR), leading to the autophosphorylation of tyrosine residues in the receptors’ cytoplasmic region. PI3K can then subsequently activate AKT, which further activates downstream effector molecules, mainly mammalian target of rapamycin (mTOR). PTEN antagonizes the action of PI3K [147]. Mutations in *PI3K*, *AKT*, and *PTEN* can lead to the abnormal activation of this pathway, which is observed in approximately 70% of BC and OC cases. *PIK3CA* and *PTEN* mutations are the most common and subtype dependent in both BC and OC. The highest percentage of *PIK3CA* mutations occurs in hormone receptor-positive (HR+)/HER2− subtypes of BC and endometriosis-associated OC (endometrioid ovarian carcinoma (EnOC) and ovarian clear-cell carcinoma (OCCC)) while the lowest percentage is found in TNBC and HGSOC [148,149]. In addition, *PTEN* mutations are more frequently associated with more aggressive types of BC and OC [39,150], although they can coexist [151]. This aberrant pathway activation contributes to resistance to chemotherapy and poor prognosis [152,153,154,155,156,157]. *PTEN* mutations or inhibition contribute to activation of the EGFR/PI3K/AKT pathway, leading to increased mTOR activation, induction of EMT, enrichment of the stem-like cell fraction, tumorigenicity, and chemoresistance [155,158,159]. Inhibition of the PI3K/AKT/mTOR pathway reduces the survival and stem-like phenotype of BCSC and OCSC [160]. The inhibition of mTOR, a signaling pathway important for maintaining the stem-like properties of cancer cells [158], negatively influences the expression of ALDH1 in CSC [161], while inhibition by metformin (an adenosine monophosphate (AMP)-activated protein kinase (AMPK) activator) leads to the suppression of proliferation in BCSC and OCSC [162,163]. The overexpression of steroid receptor coactivator 3 (SRC-3) and tumor necrosis factor (TNF)-receptor-associated factor 4 (TRAF4), activators of PI3K/AKT signaling, which is associated with a more aggressive phenotype, enrichment of CSC population, and therapy resistance, is frequently found in BC and OC [164,165,166].

**NOTCH signaling** is also associated with stemness maintenance, which is important in adult organisms and pathologies such as cancer. NOTCH1–4 is a family of transmembrane protein receptors that can bind five ligands—Jagged canonical Notch ligands 1 and 2 (JAG1, JAG2) and Delta-like canonical Notch ligands 1, 3, and 4 (DLL1, DLL3, and DLL4). When a ligand interacts with the receptor, it triggers proteolytic cleavages of the receptor by ADAM metallopeptidase domain 10 (ADAM10) and γ-secretase enzymes. This cleavage releases the intracellular portion of the NOTCH receptor, which then initiates the downstream signaling and transcription of target genes [167,168]. Hyperactivation of NOTCH signaling is crucial for the maintenance of CSC, invasiveness, and resistance to therapy in BC and OC [169,170,171]. NOTCH activity is positively correlated with ALDH1 expression in both OC [172] and BC, and NOTCH downregulation inhibits growth and induces apoptosis in ALDH1+cells [173]. Hypoxic TME induces JAG1 and activates NOTCH signaling, promoting cell proliferation, migration, invasion, chemoresistance, and an increase in the percentage of BCSC and OCSC [171,174,175]. RAS can activate NOTCH1 and upregulate DLL1, and its overexpression is correlated with the upregulation of NOTCH1 in BC [144,176]. NOTCH-positive BC cells have a higher possibility for tumor initiation as NOTCH is implicated in the transition of healthy stem cells to CSC [174,177]. Targeting the NOTCH signaling pathway with antibodies or inhibitors is shown to be an effective way of reducing the population of CD44+/CD24low BCSC, causing a decrease in mammosphere formation and brain metastasis in TNBC [178,179]. It is also effective for OCSC [180]. For example, treatment with γ-secretase inhibitor 1 (GSI), a NOTCH pathway inhibitor, has shown promise in eradicating BCSC and OCSC [181,182].

**The Hedgehog–GLI (HH-GLI) pathway** plays a major role in embryogenesis. Although its activity is reduced in the adult stage, it is highly required for the maintenance of the stem cell population, tissue repair, and regeneration [183]. In cancer, it is crucial for the regulation of stem-like genes [184] and contributes to the proliferation, drug resistance, and metastasis of CSC [185]. The pathway is activated when HH ligands bind to Patched 1 (PTCH1), derepressing Smoothened (SMO) and activating an intracellular cascade that leads to the activation of three GLI transcription factors, which are the final effectors of the pathway [186]. In OC, the overexpression of PTCH1 and GLI1 correlates with poor prognosis and survival [187]. The HH-GLI signaling was also shown to activate stemness-related transcription factors, such as *NANOG*, *OCT4*, *SOX2*, and *BMI1* in gliomas [188], while its inhibition reduces NANOG expression, decreases cell proliferation and colony formation, and abolishes cisplatin resistance in OC cells [189]. Ray et al. found that, compared to normal immortalized epithelial cells, OC cells exhibited increased GLI1 expression, spheroid-forming ability, and CSC traits (self-renewal, differentiation, and chemoresistance) [190]. The pathway is upregulated in TNBC and Luminal B BC and is associated with stem-like cell expansion [191] and vascular endothelial growth factor (VEGF)-independent angiogenesis [185]. The pathway was found to be more active in less differentiated cells, such as CD44+/CD24− cells [144,191]. As in OC, its activation is positively correlated with the expression of NANOG and OCT4 [185]. The HH signaling pathway activation contributes to enrichment of the CD44+/CD24− population [192] and chemoresistance associated with BCSC [144], while its dysregulation, observed as the overexpression of Sonic hedgehog (*SHH*), Desert hedgehog (*DHH*), and *GLI1*, is correlated with a worse prognosis and more advanced stages of BC [192,193]. The knockdown of *GLI2* can decrease ALDH activity and influence mammosphere growth, while its overexpression has a positive effect on mammosphere growth [194]. Inhibition of the pathway using inhibitors such as cyclopamine has shown negative effects on tumor progression in BC and OC mouse models [191,195]. In addition, inhibition with a GLI protein antagonist, GANT-61, affects cellular growth, stemness, and migration in OC by suppressing its target genes transcription [196] and inducing apoptosis and decreased BCSC population in TNBC cell lines [197].

**The JAK/STAT pathway** regulates cell proliferation, differentiation, apoptosis, and immune response by being stimulated mainly by growth factors (EGF, IGF, and FGF) and cytokines (IL6, IL5, IL8, IL10, and IL23) [198,199]. There are three Janus Kinase (JAK) protein (1–3) and seven signal transducer and activator of transcription (STAT) proteins (0–6). Ligands binding to membrane receptors initiate JAK transphosphorylation, resulting in the phosphorylation of tyrosine residues on the receptor. These phosphorylated sites serve as docking sites for STAT proteins. JAK phosphorylates STATs, which then dissociate from the receptor and form dimers. These dimers subsequently translocate into the nucleus, where they regulate the transcription of specific target genes [200]. The JAK/STAT pathway has been implicated in CSC biology in several cancer types, including BC and OC, where it plays a role in cancer initiation, progression, multidrug resistance, and metastasis. In OC, inhibition of the JAK/STAT pathway resulted in a loss of CSC-like markers and CA125 expression [201]. Moreover, increased phosphorylation of STAT3 and expression of STAT3 target genes, *NANOG* and *C-MYC* in CD24+ OCSC, when inhibited with JAK2, induces cytotoxicity in CD24+ cells, reduces tumor metastasis, and prolongs overall survival [202]. Ruan and colleagues demonstrated that the inhibition of JAK/STAT signaling attenuated the tumor-feeding effects caused by the upregulation of OCT4 in OCSCs [203]. Moreover, IL23 has been shown to maintain the tumorigenic potential of CD133+ OCSCs in vivo and mediate their ability to self-renew through activation of STAT3 [204]. Overexpression of STAT3 is found in more than 40% of BC, mainly in TNBC [144]. Similar to OC, activation of JAK/STAT is important for the occurrence of BCSC, such as an increase in the CD44+/CD24low BCSC population observed in hypoxia, leading to the upregulation of genes that influence angiogenesis, proliferation, and EMT, further promoting BC development and chemoresistance [144,199]. STAT3 upregulates matrix metalloproteinases (MMP), especially MMP-2 and MMP-9, which promote cancer invasion and metastasis [199]. ROS- and IL6-induced activation of STAT3 promotes BCSC occurrence, BC progression, and inflammation [199]. In addition, IL6- and IL10-induced activation of STAT3 is involved in BC metastasis to the liver, bones, and lungs [205]. STAT3 interacts with CD44, nuclear factor kappa B (NF-κB), and catalytic subunit of telomerase (hTERT) to promote the BCSC phenotype [199]. BCSC with ALDH+ and ALDH+/CD44+/CD24− phenotype were associated with the activation of JAK/STAT signaling, and the use of STAT3 inhibitors (Stattic or LLL12) reduced the growth and invasiveness of BCSC while inducing apoptosis [206,207]. In both BCSC and OCSC, activation of JAK/STAT signaling promotes macrophage M2 polarization and cancer cell invasion [208], which negatively correlates with patient survival [209]. 

**The transforming growth factor-β (TGFβ) signaling pathway** plays a crucial role in maintaining normal tissue homeostasis, but its dysregulation is implicated in tumorigenesis, including BC and OC. TGFß binds to the specific receptor complex consisting of the TGF-β type II receptor (TGFβRII) and the TGF-β type I receptor (TGFβRI). Ligand binding with TGFβRII leads to the receptor phosphorylation forming a binding site for TGFβRI, in which phosphorylation triggers the activation of the receptor-activated suppressor of mothers against decapentaplegic (SMAD) proteins that translocate to the nucleus and regulate transcription of targeted genes. TGF-beta signaling can also activate non-SMAD pathways, which contribute to additional cellular responses [210]. Activation of TGFß signaling and cooperation with the TNFα and WNT pathways contributes to increases in BCSC and OCSC populations, tumorigenicity, and chemoresistance through regulation of EMT [113,211,212,213,214,215,216]. By inducing the expression of TTG2, TGFß promotes EMT, spheroid formation, and metastasis in OC, while *TTG2* knockdown decreases the number of cells with a CSC phenotype (CD44+ CD117+) [217]. Silencing of SNAIL, a known TGFß target, reverses stemness properties and inhibits tumor growth in OCSC [137]. Inhibition of TGFß has also been shown to be effective in restoring sensitivity to cisplatin in OCSC [218]. In addition, the paclitaxel-induced increase in TGFβ and IL8 in TNBC, which led to an increase in BCSC number, could be reversed by inhibition of the TGF-β type II receptor [219]. In both BC and OC, the TGFß signaling pathway influences TME, especially cancer-associated fibroblasts (CAF), to promote tumor growth. It transforms normal fibroblasts into CAF [220], which continue to produce extracellular matrix proteins that support the microenvironment niche for CSC growth [221]. In addition, platelet-derived TGFβ can activate the TGFβ/SMAD and NF-κB signaling pathways, which can stimulate cancer cells to metastasize, while inhibition of TGFβ can reduce the lung metastasis of BC [222]. Treatment of OC cell line SKOV-3 with SB525334, an inhibitor of TGFβRI, showed promising results in targeting CSC with high ALDH expression. The self-renewal ability of CSC and in vitro tumorigenicity were reduced when treated with SB525334, which indicates the possibility of CSC-targeted treatments for OCSC and other CSC with high ALDH expression, such as BCSC [223].

**The NF-κB signaling pathway** plays a crucial role in cell functions and processes such as inflammation, angiogenesis, differentiation, stemness, and metastasis in BC and OC [224,225]. The activity of NF-κB is regulated by the inhibitor of κB (IκB) proteins. These IκB proteins act as inhibitors, preventing NF-κB from entering the nucleus. The IκB kinase (IKK) complex is responsible for regulating the IκB. When specific signals, such as pro-inflammatory cytokines, are recognized, the IKK complex is activated. This activation leads to the degradation of IκBs and the subsequent release of NF-κB. As a result, NF-κB translocates to the nucleus and controls the transcription of targeted genes [226]. Constitutive activation of the NF-κB pathway is observed in BCSC and OCSC. Activation of NF-κB in TNBC correlates with an increase in the CD24−/CD44+/EpCAM+ fraction and ALDH+ fraction of BCSC [227], while inhibition of NF-κB decreases CD44 expression, invasiveness, and proliferation of TNBC cells [228]. Similarly, NF-κB was found to be constitutively active in CD44+ OCSC and to promote chemoresistance, spheroid formation, and self-renewal/proliferation of OCSC [229]. Inhibition of NF-κB restored the response to platinum therapy in OCSC [230] and specifically decreased the expression of CD44 and genes involved in OCSC self-renewal/proliferation [231]. Together with the JAK2/STAT signaling, NF-κB seems to be crucial for OCSC self-differentiation into endothelial cells and promotion of angiogenesis [232]. Moreover, NF-κB signaling is involved in the interaction of OCSC with the TME. Namely, OCSC release cytokines to promote M2 macrophages’ recruitment and support of the cancer stem niche [79,130,204,209]. Inhibition of the NF-κB signaling pathway with a dominant-negative version of IkBα in OC reduced the number of CD44+ OCSC and the expression of the stemness genes [233]. In BC, inhibitors of NF-κB signaling (parthenolide, pyrrolidinedithiocarbamate, and its analog diethyldithiocarbamate) inhibited MCF7 mammosphere growth, suggesting its preferential targeting of BCSC [234]. 

**The Hippo signaling pathway** regulates a wide range of biological processes, including cell survival, differentiation, cell proliferation, and tissue homeostasis [235]. The Hippo pathway involves a kinase cascade comprising mammalian Ste20-like kinases 1/2 (MST1/2) and large tumor suppressor 1/2 (LATS1/2), along with downstream effectors, namely, the transcriptional coactivators Yes-associated protein (YAP) and the transcriptional coactivator with PDZ-binding motif (TAZ). Various cellular factors such as cell density, cell polarity, soluble factors, stress signals, and mechanical cues activate the pathway [236]. YAP, the major target of the Hippo pathway, is a known oncogene that, when suppressed, restores sensitivity to platinum-based therapy in OC and leads to decreased stemness properties in BC and OC [237,238,239]. The self-renewal and chemoresistance of BCSC and OCSC depend on the activity of the YAP and Hippo downstream coactivator TEAD [239,240]. In BC, hyperactivated YAP leads to the formation of BCSC, while TAZ is responsible for maintaining stem-like properties in CD44+/CD24− BCSC [241]. Detachment of cells from the extracellular matrix (ECM) leads to YAP/TAZ inhibition, resulting in anoikis, while dysregulation of HIPPO/YAP/TAZ leads to anoikis resistance and EMT [241]. RICH1, a Rho GTPase-activating protein, increases the sensitivity of BC to chemotherapeutics and inhibits stem-cell-like properties by inhibiting nuclear translocation of YAP/TAZ. This emphasizes the possibility of CSC-targeted therapies through manipulation of the HIPPO signaling pathway [242]. Indeed, verteporfin has been shown to be a suppressor of the YAP–TEAD complex with anticancer properties. OC cell lines treated with verteporfin showed decreased in vitro migration and reduced tumor growth in xenograft mice [243].

**The nuclear factor erythroid 2-related factor 2 (NRF2) signaling pathway** is often associated with the enrichment of BCSC and OCSC, leading to therapy resistance in BC and OC. NRF2 is a transcriptional factor that regulates the expression of more than 250 genes. It is repressed in a complex with Kelch-like ECH-associated protein 1 (KEAP1) and subjected to ubiquitination and proteasomal degradation. However, NRF2 regulation/activation can include other repressors or competitors. Oxidative stress activates this pathway by blocking the binding of NRF2 with KEAP1, which fosters its translocation to the nucleus and activation of target genes with antioxidant-responsive elements (ARE) in their promoter [244]. Radioresistance and chemoresistance in BC are linked to the augmentation of ALDH+CD44+ BCSC [245], but mainly to ALDH+ BCSC [71]. Similarly, in OC, chemoresistance is linked to OCSC expressing high levels of ALDH1 and not CD24, CD117, or CD133 [246]. Stemness properties, such as drug resistance, colony/sphere formation, tumor growth, and high stemness marker expression, of both BCSC and OCSC are regulated by the p62/NRF2 axis. The molecular mechanism involves the activation of NRF2 through the p62-associated pathway [247], regulated by ALDH1A1 in OC [248] and CD44 in BC [249]. Furthermore, resistance to radiotherapy in BC is associated with activation of the NRF2 pathway, the enrichment of ALDH+ BCSC, and the promotion of EMT. The activation of the NRF2 signaling pathway in BC can be mediated by reactive oxygen species (ROS) [250] or can occur via the silencing of KEAP1 by miR200a [251]. GSK-3β and BTB Domain And CNC Homolog 1 (BACH1) seem not to be involved in the radiation-induced activation of NRF2 [251]. ML385-mediated inhibition of NRF2 has been found to enhance the sensitivity of BCSC to ionizing radiation, while the activation of NRF2 through sulforaphane diminishes this effect [250]. Furthermore, the inhibition of NRF2 in ALDH+ OC cells and CD44+ BC cells has demonstrated a reduction in CSC properties, including chemoresistance, tumor growth, and spheroid formation [233]. 

These findings highlight the significance of targeting pathways involved in CSC biology as a potential therapeutic strategy to overcome therapy resistance in breast and ovarian cancers.

## 5. BCSC- and OCSC-Microenvironment Communication

The microenvironment of both BC and OC includes several cell types, including stromal cells, adipocytes, carcinoma-associated mesenchymal stem cells (CA-MSC), and immune cells (Figure 2). All these cell types create a communication network where many signaling molecules are released and maintain a specific local microenvironment. Single-cell RNA-sequencing analysis of OC revealed several major clusters within the tumor: epithelial, mesenchyme, macrophage, differentiation, T cell, B cell, and endothelium clusters. Within the epithelial cluster, a specific carcinoembryonic cluster was identified, PEG10+, which showed high similarities with embryonic cells, and is associated with carcinogenesis [252].

### 5.1. The Connection between Adipocyte Populations and BCSC and OCSC

Obesity is a known risk factor for many cancer types, including OC and BC [253,254]. A recent single-cell and spatial transcriptomic meta-analysis of human white adipose tissue revealed the presence of over 60 distinct cell populations within four major cell types: adipocytes (~20%), fibroblast and adipogenic progenitors (~40%), vascular cells (~15%), and immune cells (~20%) [255]. Adipocytes, which are common to both OC and BC, contribute greatly to the carcinogenesis. High visceral fat-to-muscle ratio is a predictor of worse overall survival for OC patients [256]. Adipocytes secrete adipokines, growth factors, and hormones which are supportive for tumor growth. For example, resistin induces stemness and EMT of EOC cells, and triggers angiogenesis through VEGF and MMP-2 [257] Additionally, adipocytes via leptin signaling increase the expression of stem markers such as SOX2 and NOTCH2 and contribute to the enrichment of BCSC [258]. Adipocytes also induce lipid metabolism in cancer cells, and co-culturing cancer cells with adipocytes leads to an increase in β-oxidation, ROS, and lipid peroxidation [259,260]. Additionally, they induce nitrogen oxide (NO) synthesis in OC cells, which then use arginine from the TME and secrete citrulline, further inducing adipogenesis [261]. OC cells, on the other hand, also affect the behavior of adipocytes. WNT signaling components secreted from OC cells can induce defattening of adipocytes and their conversion into **adipocyte-derived fibroblasts (ADF)**, which demonstrate CAF-like characteristics. This morphological change is followed by changes in reduced expression of adipose markers and an increase in fibroblast markers, creating a subpopulation of cells at the invasive front of the tumor. These cells can migrate to the tumor center and contribute to tumor invasiveness and migratory capacity [262]. Even more, there are breast adipose tissue-derived mesenchymal stromal/stem cells (bASC), which can de-differentiate into different CAF and stimulate proliferation, migration, chemoresistance, and stemness of BC cells [263,264]. ADF cocultured with OC cell lines induce their proliferation and migratory ability, and induce their fatty acid uptake and lipid accumulation [265,266]. However, there is also evidence of an inhibitory effect of OC on adipocyte differentiation through the ECM protein secreted protein acidic and rich in cysteine (SPARC), which is responsible for maintaining tissue homeostasis [267]. 

**Adipose-derived stem cells (ADSC)** exhibit features of mesenchymal stromal cells, and can differentiate into various cell types: adipocytes, osteoblasts, chondrocytes, or myocytes. ADSC-conditioned medium increases migration and invasion of OC cell lines and xenografts, and promotes EMT through the TGF-β pathway [268,269]. ADSC also influence the sphere formation and tumor initiation of BC, along with increased expression of stem markers ALDH1A1 and ABCG2 [270]. Conversely, exosomes from OC and BC cells can direct the differentiation of ADSC into a myofibroblast-like phenotype [271,272]. To understand the role of adipocytes in tumor development and progression, it has to be noted that white adipose tissue (WAT) is an endocrine organ that also has a metabolic function [273,274]. WAT secretes inflammatory cytokines such as IL1, IL6, IL10 and TNFα, but also TGFβ, monocyte chemoattractive protein-1 (MCP-1), angiogenic proteins such as C-X-C motif chemokine 5 (CXCL5) and VEGF, but also adipokines such as leptin, resistin, and adiponectin (reviewed in [273]). These cyto/chemo/adipokines play an important role in determining the fate of tumors in their vicinity. IL6, a pro-inflammatory cytokine secreted locally and systematically by adipocytes, is significant for tumor development, as high plasma levels correlate with poorer overall survival of BC patients [262,275]. The secretion of IL6 is caused by tumor stimulation of adipocytes, as shown by co-cultivation of murine adipocytes with human BC cell line ZR75.1 [262]. In response to IL6, CD44+CD27− BC cells specifically activate the JAK2/STAT3 pathway, whereas NF-κB signaling is not as specific for these basal-like BC cells [276]. Yet, in HER2+ BC, IL6 can promote stemness and metastases via NF-κB/STAT3 activation [277]. In OC, IL6 can activate the JAK/STAT3 signaling pathway, leading to tumor growth, metastasis, and EMT [278]. In addition, IL6 triggers NOTCH3-dependent upregulation of genes promoting malignant features of HR+ MCF7 spheroids [279]. TNFα, also a pro-inflammatory cytokine, can trigger the stemness phenotype and drug resistance of basal-like BC cells through activation of the KLF5/Ephrin type-A receptor 2 (EphA2) axis [280]. Furthermore, **cancer-associated adipocytes (CAA)** can trigger increased expression of pro-inflammatory cytokines *IL6* and *IL8* in co-cultures with TNBC cells MDA-MB-231 [281]. TNFα also influences the expression of CD44 in OC through the activation of JNK, where it increases CD44 expression; this activation of JNK results in a more aggressive phenotype [282]. TNFα production in OC leads to increases in IL6, VEGF, C-C motif ligand 2 (CCL2), and C-X-C motif chemokine ligand 12 (CXCL12), and lower levels of TNF-α lead to a reduction in tumor growth [283].

### 5.2. The Effect of Immune Cells on BCSC and OCSC

The tumor-promoting microenvironment is generated by stimulating angiogenesis (activation of PI3K pathway via VEGF and EGF), regulating adhesion (changes in composition of integrins), and inflammation (pro-inflammatory cytokines IL6 and IL8 and increased neutrophil to lymphocyte ratio). Bioactive lipids such as lysophosphatidic acid (LPA) and arachidonic acid (AA) are elevated in malignant ascites and contribute to immunosuppression [284]. Many immune cell subtypes are found accumulated within the ascites, including M2 polarized TAM, myeloid-derived suppressor cells (MDSC), CD4+ and CD8+ T cells, Tregs, DC2 dendritic cells, and cytokine-producing natural killer (NK) cells, while cytotoxic NK cells are reduced [55]. Similar trends can be seen in BC metastasis, where the microenvironment plays a crucial role in the spread of the tumor, and many immune cells promote immunosuppression and tumor progression, including M2 macrophages, neutrophils, CD8+, CD4+, and Tregs [285,286]. Extracellular vesicles (EV) mediate the crosstalk between tumor cells and their microenvironment. EV can trigger immunosuppression by arresting T cells, inducing TAM polarization into M2 subtype, and inducing IL6. Additionally, EV can induce transformation of host stromal cells into CAF [287]. In BC, EV promote lung metastasis [288], increase migration and invasion of BC cells, and enhance EMT [289], while inhibition of EV leads to decreased tumor growth and lung metastasis [290]. 

Classically activated macrophages, also known as M1, are pro-inflammatory, while M2 macrophages are further divided into different subgroups (M2a, M2b, M2c, and M2d); furthermore, they have immunosuppressive properties along with tumor-promoting properties, so they are often called **tumor-associated macrophages (TAM)** [291]. TAM are one of the most prevalent types of immune cells found in BC and affect BCSC properties. BCSC interacting with macrophages initiates their tumor-promoting function, and these TAM produce high levels of IL6, which in turn initiates activation of the STAT3 signaling pathway and consequent enrichment of BCSC [292]. TAM also interact with OCSC and have been shown to correlate with poor overall patient survival, chemoresistance, and promotion of metastasis in OC [293]. OCSC can influence M2 macrophages and promote immunosuppression, while at the same time M2 macrophages can promote stemness of OCSC [139].

One of the most important components of the immune system in relation to cancer cells is the population of **cytotoxic CD8+ T cells**, which are heavily involved in BCSC and OCSC function. Since the function of cytotoxic CD8+ T cells is to kill abnormal cells, they are of great importance in the regulation of cancer growth. At the onset of cancer growth, CD8+ cells are responsible for killing cancer cells, and their presence in cancer tissue correlates positively with better overall survival of BC patients [294]. However, when tumors advance, the role of CD8+ T cells changes—they can no longer destroy tumor cells and instead promote the growth of the tumor rather than its eradication. BCSC, OCSC, and stromal cells can secrete large quantities of TGF-ß, a well-known CD8+ cell inhibitor [295,296]. Other than TGF-ß, BCSC can express high levels of Programmed cell death ligand 1 (PD-L1). This is especially important because CD8+ T cells express inhibitory receptor Programmed death-1 (PD-1), and through their interaction, BCSC along with OCSC can inactivate CD8+ T cells, further improving the chances for tumor development [297,298]. TNBC cells, which exhibit high levels of PD-L1, form more mammospheres than cells with a lower expression of PD-L1 [299]. Furthermore, the high expression of PD-L1 is positively correlated with the expression of other stem markers in BCSC, such as CD44 and ALDH [299]. Interaction between MCF7 cells and CD8+ T cells that have been treated with Concanamycin A, so they are unable to lysate tumor cells, has been shown to increase the population of CD44+/CD24− cells with stem properties [300]. Even though the levels of CD8+ T cells correlate with better survival of patients with OC, this effect seems to be dependent on the presence of CD4+ tumor-infiltrating T cells (TIL), where CD25+FOXP3+ regulatory T cells mitigate the effect of CD8+ T cells [301]. This polarization of CD4+ cells plays a part in tumor formation and growth. CD4+ T helper cells play an important role in cancer inhibition and destruction, while regulatory T cells have been shown to promote cancer. The expression of SOX2 in BCSC recruits the regulatory T cells through secretion of chemokine (C-C motif) ligand 1 (CCL-1), where Tregs further promote CSC properties [302].

### 5.3. Cancer-Associated Fibroblasts and Mesenchymal Stem Cells

**CAF** are generally found in close physical relation to a tumor, and they are characterized by the expression of fibroblast activation protein (FAP) and actin smooth muscle (αSMA) [303]. BCSC express high levels of SHH, a ligand of the HH signaling pathway that can in turn activate the HH signaling pathway in CAF via paracrine activation, which in turn increase stem properties of BCSC [304]. CAF are involved in ECM remodeling, regulation of immune cells, and promotion of stem characteristics of BCSC and OCSC [305,306]. ECM remodeling by replacing collagen type 4 with collagen type 1 and 3 [307] or by MMP influences angiogenesis and metastasis [305]. CAF secrete chemokines and cytokines that regulate regulatory T cells (Treg)-mediated immunosuppression, cytotoxic T cell localization, and macrophage differentiation into TAM [305]. CAF can induce OCSC through IL8 signaling and activation of the NOTCH signaling pathway [306]. OC cells cocultured with CAF show increased chemoresistance and increased number of OCSC. An increase in OCSC is obtained through WNT signaling, and its inhibition reversed the effect of CAF on OCSC [308]. Through para- and autocrine interaction, CAF can influence OC angiogenesis, metastasis, and proliferation mediated by various signaling pathways, such as AKT/mTOR, TGF-β, and NF-κB [309]. These data indicate the important role of CAF in the progression of BC and OC through its interaction with CSC.

**CA-MSC** are another crucial component of the OC TME. They are defined by the expression of CD44, CD73, and CD90 markers, and can differentiate into adipose tissue, cartilage, or bone in vitro [310]. CA-MSC originate from local abdominal tissue and are induced by hypoxia [311]. Hypoxia can induce a quiescent state but also increase proliferation, invasion, and sphere-formation capabilities, as well as increased stem-like properties of OC cells [175,312]. Hypoxia can also influence the expression of BCSC markers, such as CD133, where low-oxygen status leads to an increase in CD133 expression [76]. One of the proposed mechanisms is the hypoxia-induced upregulation of NOTCH signaling, leading to the upregulation of SOX2 and ALDH [175]. Another mechanism proposes the accumulation of HIF-1α and HIF-2α in OC cells upon ammonia treatment [313]. Hypoxia also induces the expression of EMT hallmark proteins such as expression of *SLUG* and *SNAIL* in BC cells [314]. Sirtuin 1 (SIRT1) is a downstream target of HIF-1α and has been proposed as a potential therapeutic target for OC [315], while SIRT1 increases chemoresistance and metastasis, and its overexpression is connected to BC growth [316,317].

## 6. Cancer Immunogenicity and Evasion of Immunosurveillance of OC and BC

The immune system plays a crucial role in shaping the development and progression of cancer, and according to the “cancer immunoediting” hypothesis it comprises three phases: elimination, equilibrium, and escape. During the elimination phase, the immune system (innate and adaptive) detects and eliminates tumor cells that display foreign antigens on their surface. Yet, some tumor cells develop mechanisms to evade or suppress immune responses, leading to the equilibrium phase. In the equilibrium phase, the immune system (mainly T helper type 1 (Th1) cells, cytotoxic T cells, and cytokines of type 1 immunity (IL12, IL2, interferon gamma (IFN-γ)) keeps tumor cells dormant or growing slowly. The escape phase occurs when tumor cells develop genetic instability and evade immune detection, resulting in rapid growth and metastasis. It is when the tumor becomes clinically visible. This phase is aided by a proangiogenic microenvironment and the suppression of the immune response. This includes Treg cells, myeloid-derived suppressor cells, type 2 macrophages, IL10, TGF-ß, VEGF, 2,3-dioxygenase (IDO), arginase-1, and cyclooxygenase-2 (COX-2) as well as the overexpression of pro-survival proteins and inefficient presentation of tumor-associated antigens [318]. 

While BC was firstly perceived as an immunogenically “cold” tumor, OC is considered an immunogenic tumor, as it expresses so-called tumor-associated antigens (TAA) that can be recognized by the immune system. These antigens are produced by the tumor cell, phagocytosed, and processed by the dendritic cells, which then expose them on the major histocompatibility complex (MHC) class molecules on their surface. These are recognized by the T cells and trigger the immune response of the host. TAA identified in OC include HER2/neu, CA125, FR-α, CA153, HE-4, and others [319]. Today, BC is also considered an immunogenic tumor due to its highly dynamic immune heterogeneity. Indeed, immune cells are present in the tumor and TME but are exposed to highly immunosuppressive environment. TAA identified in BC are HER2, Mucin 1 (MUC1), carcinoembryonic antigen (CEA), NY-ESO-1, melanoma-associated antigens (MAGE), brachyury, cMET (MET receptor tyrosine kinase), and mesothelin [320]. HER2 is a tyrosine kinase receptor of the EGF family overexpressed in 15–30% of invasive BC and 20–30% of OC. Its overexpression correlates with worse survival of BC and OC patients and more aggressive disease [321,322]. Interestingly, the same HER2/neu sequence is recognized by HLA-A2-restricted tumor-specific cytotoxic T lymphocytes found in both BC and OC [323]. This suggests that targeting this sequence with immunotherapy could potentially be effective in both types of cancer. 

NK cells act cytotoxically on the cells that lose MHC class molecules. In OC, CD24+ cell populations isolated from OC cell lines were more susceptible to NK cell cytotoxicity as they show loss of MHC class I molecules on their surface, and upregulation of NKG2D receptor, which is responsible for initiation of NK killing mechanism. On the other hand, CD24+ population showed increased resistance to cisplatin and doxorubicin compared to CD24− population [324]. In fact, a study comparing seven OC cell lines according to their level of differentiation and CD44 expression concluded that poorly differentiated lines, with high CD44 and low MHC class I expression, are effectively eliminated by NK cells, and this effectiveness drops with the increasing differentiation level and MHC class I expression [325]. However, the ascitic fluid of advanced OC suppresses the effect of NK cells [326]. In BC, NK cells have been shown to reduce tumor growth in vivo [327] and specifically kill CD44+CD24low/- BCSC from an HR+ cell line when activated with IL2 and IL15 [328]. Conversely, analysis of radioresistant TNBC showed enrichment in CD44+CD24low/- BCSC subpopulations that evade the cytotoxic activity of NK cells. Although NK cells invade the tumor enriched with CD44+CD24low/- BCSC, their activity is impaired due to altered expression of ligands on BCSC. Specifically, the expression of HLA-E, the ligand for the inhibitory NKG2A receptor, is increased and the expression of MICA/B, the ligands for the activating NKG2D receptor, is decreased. Cleavage by ADAM10 [329] or downregulation by the oncogenic miR20a [330] contribute to decrease in MICA/B expression and subsequent deficiency in NKG2D recognition. Moreover, the accumulation of MDSC that negatively regulate NK cell activation and function contributes to the evasion of CD44+CD24low/- BCSC from NK cells and BC progression as well [329]. In an early phase clinical trial, the infusion of activated allogeneic NK cells in BC and OC patients resulted in transient donor chimerism, but there was no notable increase in the number of transfused NK cells [331].

In both, OC and BC, noncanonical activation of the TGFβ pathway in CD8+ T cells upregulates CD103 and induces secretion of CXCL13. These CXCL13+CD103+CD8+ TIL subpopulations promote migration of B cells to the tumor and the formation of tertiary lymphoid structures (TLS) and are associated with better prognosis [332]. In addition, a large-scale meta-analysis identifying the immunogenic cell death (ICD)-metagene expression signature in BC, OC, and lung cancer according to patients’ better prognoses reveals a highly cancer-type-specific prognostic impact, albeit with some similarities between BC and OC. High expression of the BC-specific ICD-derived metagene signature (*TNF/CXCR3/P2RX7/CASP1/NLRP3/IL1B/LY96/CD4+/CD8+A/CD8+B/PRF1/IFNG/IL17A/IL17RA*) is associated with prolonged overall survival (OS) in BC and OC patients but not in lung cancer patients. Similarly, high expression of the OC-specific ICD-derived metagene signature (*CALR/PIK3CA/TNF/IFNA/IFNB1/CXCR3/P2RX7/CASP1/IL1B/TLR4/CD4+/PRF1/IFNG/IL17A/IL17RA*) is associated with prolonged OS in OC patients and to some extent with prolonged OS in BC patients, whereas high expression of the lung-cancer-specific ICD-derived metagene signature is associated with prolonged OS only in lung cancer patients. This suggests that immune or inflammatory responses may operate in a more unified manner in BC and OC [333]. 

OC tissues express PD-L1 in approximately 50% of cases, and its expression correlates with high CD8 expression and moderate CD4 expression. Interestingly, the expression of PD-L1 colocalizes with the stem cell markers CD44 and LGR5 [334]. In BC, the expression of PD-L1 is three-fold higher in BCSC than in more differentiated cancer cells [335] and correlates with immune evasion [336]. Moreover, PD-L1 can be found in various immune cells, including macrophages, CD4+, FOXP3+, and CD8+ T cells within the BC TME, and higher levels are associated with better prognosis in TNBC [337]. In OC, a higher percentage of PD-L1 positive cells was found in the OC infiltrating cells compared to peripheral blood, and the highest in peritoneal fluid [338]. However, a meta-analysis of PD-L1 expression in OC revealed that PD-L1 does not have prognostic value in OC [339], in contrast to BC where a similar meta-analysis identified high PD-L1 expression as a negative prognostic factor [340]. However, PD-L1 upregulation has been associated with the development of carboplatin resistance in OC [341]. CD24, one of the OCSC markers, is expressed at higher levels than PD-L1 in OC, and interacts with Siglec-10 expressed on TAM, leading to anti-phagocytic activity of TAM and immune evasion. The same interaction has also been demonstrated for TNBC, and the authors propose anti-CD24 immunotherapy as a mechanism to promote immune response in OC and TNBC patients [342]. 

Cancer progression, metastasis, recurrence, and resistance occurs due to the BCSC and OCSC escape from immune surveillance [343,344]. They achieve immune evasion through several mechanisms. In OC, activation of the NF-κB signaling pathway directs the macrophages towards M2 polarization, which is associated with anti-inflammatory and immunosuppressive processes [209]. M2 macrophages express CD39 and CD73, which are responsible for the conversion of ATP to adenosine, which additionally pushes the macrophages towards the M2 phenotype, and suppresses CD4+ T, CD8+ T, and NK cells, thus generating a self-amplifying mechanism for immune evasion [345]. CD163+ TAM are more frequent in the ascites of OC patients compared to the peripheral blood of the same patients, and exosomes derived from these cells promote adhesion and migration of EOC cells [346]. Interaction of BCSC with macrophages initiates their tumor-promoting function, and these TAM produce high levels of IL6 which in turn initiates activation of the STAT3 signaling pathway and consequent enrichment of BCSC. These BCSC are also enriched in SOX2, NANOG, Stem cells antigen-1 (SCA-1), and OCT4 [292]. CSC can produce chemokines, such as CCL2, CCL3, CCL5, and CCL8, to recruit monocytes, thus influencing monocyte migration [347].

Regulatory T cells CD4+CD25+CD3+ are abundant in the tumor mass and the malignant ascites of OC patients with later stage disease, and they suppress the production of IFN-γ and IL2 and subsequent T cell activation [348]. Aging contributes to the reduced function of the immune system, as aged CD4+ TIL and aged CD20+ B cells are less active than young ones [349]. In fact, ascites-derived T cells show poor proliferation after stimulation with αCD3/28 beads and are unresponsive to IL2. The components responsible for this effect were shown to be the lipids within the ascites [350]. Polarization of CD4+ T cells also plays a part in tumor formation and growth. CD4+ T helpers play an important role in cancer inhibition and destruction, while regulatory T cells have been shown to promote cancer. Expression of SOX2 in BCSC recruits the CD4+ regulatory T cells through secretion of CCL1, where Tregs further promote cancer cell stem properties [302].

## 7. Experimental Models for Distinguishing CSC Populations

One of the most common problems with studying CSC has been the selection of a method for the isolation and characterization of CSC. Throughout the years, several types of models and methods were used in order to distinguish CSC populations, all with their advantages and limitations. 

The use of culture models in cancer research provides a valuable tool for investigating various aspects of tumor biology, including drug response, drug resistance, TME, and stem-cell-like properties. While the predominantly used 2D cultures provide a simple and high-throughput method in this field, they lack physiological relevance due to the altered 3D architecture, cell-to-cell interactions, and behavior. Three-dimensional models, by increasing complexity, include spheroid cultures of tumor cells, spheroid co-cultures of two or more cell types, and organoids developed from patient material, as the most complex system. These models more accurately mimic the complex architecture and microenvironment of tumors, cell heterogeneity, and the presence of CSC, but their use can present challenges such as difficult imaging, higher cost, and a lack of standardization. Several reviews already discussed the advantages and disadvantages of using 3D versus 2D cultures in cancer research. They listed different 3D culture models, including spheroids grown on ultra-low attachment plates or in spinner flasks, synthetic or natural hydrogel models and 3D-printed scaffolds, microfluidic organ-on-a-chip models, and cell line or patient-derived xenografts, and discussed their applications and limitations [351,352,353,354,355,356,357,358,359,360]. The ability of 3D models to mimic the architecture and microenvironment of tumors enables the maintenance of undifferentiated states and cancer stem-cell-like properties, providing insight into the role of CSC in cancer initiation, progression, and resistance to treatment. 

Studies have shown that **3D models** exhibit increased resistance to chemotherapeutics compared to 2D cultures, and they can demonstrate the efficacy of drugs in eradicating CSC populations and reducing spheroid size. An increased resistance to chemotherapeutics was shown on spheroid models [361,362,363,364,365,366,367,368,369,370,371], scaffold-based models [372,373,374,375,376,377,378,379,380], and organs-on-a-chip [381,382,383] when compared to 2D cell cultures. Numerous studies have reported the use of various models in BC and OC research showing increased levels of cancer stem-like markers. For example, cultivating the BC cell lines in natural 3D collagen scaffolds promoted EMT and resulted in increased tumorigenicity and BCSC populations with a CD44+/CD24− phenotype and markers *OCT4*, *SOX2*, *SOX4*, *JAG1*, and *CD49f* [384]. Spheroids grown on hydrogel-based 3D scaffolds maintained a drug-resistant phenotype (CD44+/CD24−/ALDH1+) and exhibited an increased drug resistance [374]. Cultivating BC cell lines on a polycaprolactone (PCL) scaffolds in 3D culture conditions resulted in an elevated proportion of BCSC populations (CD44+/CD24−) [385], an upregulation of stem cell markers (*OCT3/4*, *SOX2*, *SOX4*, and *CD49f*) and increased invasive capability [386], and an increased metastatic potential [387], accompanied by a significant increase in the resistance to chemotherapy [388]. Omentum is one of the preferred sites of OC metastases, so 3D omentum-inspired hydrogel and a four-cell-culture 3D model using primary mesothelial cells, fibroblasts, adipocyte cells, and high-grade OC cell lines were developed to study OC metastasis and drug response of patient-derived OC cells [389,390]. In addition, single-cell-derived metastatic OC spheroids (sMOCS) from ascites showed key features of cancer stemness of the original metastasis only in 3D and not in a 2D model [391]. Recently, a more sophisticated dynamical 3D model, mimicking hydrodynamic forces OC cells experience in the peritoneal cavity, was developed [392].

**Xenograft models** have also been successfully used in BC and OC research, and both BCSC and OCSC were shown to accelerate the tumor growth in mice [393,394]. Patient-derived xenografts (PDX) have been used in therapy efficiency and chemoresistance studies and were shown to be a good model for observing the changes in CD44+/CD24− BCSC populations and ALDH1+ OCSC populations [104,179,394,395]. PDX models have been shown to exhibit clinical and molecular characteristics of primary tumors, and can provide valuable information on tumor growth, metastasis, drug efficacy, and even prognosis both in breast [396,397,398] and ovarian cancer [399]. PDX models identified the hilum region of the mouse ovary, the transitional/junction area between ovarian surface epithelium (OSE), mesothelium, and tubal epithelium, as a previously unrecognized stem cell niche of the OSE [400]. 

Furthermore, decellularizing primary BC tissues led to the development of **patient-derived scaffolds (PDS)**, which, when recellularized with BC cell lines, led to the development of stem-cell-like properties and a gene-expression profile similar to xenograft cultures [401]. In follow-up studies, the same group reported a change in BCSC markers *NANOG*, *POU5F1*, and *ABCG2*, and increased resistance to chemotherapeutics (5-fluorouracil, doxorubicin, and paclitaxel) [402] and endocrine therapies [403] in cells grown in PDS compared to 2D cultures, as well as an upregulation of CSC markers and a change of gene expression linked to prognostic features of the original cancer [404]. 

Regarding the **patient material**, the models most often used are the paraffin-embedded tissue slides and fresh or fresh frozen tissue samples (including ascites aspirates, biopsies, or solid tumors excised during surgery). They are used for the detection of gene and/or protein expression and mutation detection. For example, immunohistochemical staining and/or CSC-markers-based cell sorting revealed the abundance and CSC distribution between normal and cancer tissue at different stages and clinical outcomes [405,406,407,408].

Fresh tissues can be used to establish organoids/tumoroids in vitro, or for propagation in mice in PDX models [409]. 

## 8. CSC-Targeted Therapies

The heterogeneity of OC and BC is largely attributed to the presence of a subset of CSC, which have the ability to differentiate into multiple cancer cell types under various stimuli. Aggressive subtypes typically have a higher proportion of CSC, which can drive tumor development, progression, and resistance to therapy. 

While primary tumors treated with specific therapy may initially respond well, they often become resistant and relapse when treated again with the same compound. This resistance to therapy is partially mediated by CSC, which can evade initial treatment by persisting in a quiescent, low proliferative state, and then re-activating and repopulating the tumor mass after therapy, often adopting a more resistant phenotype. CSC are intrinsically more resistant due to their active DNA repair, increased expression of ABC transporters, ALDH1, pro-survival BCL-2 protein family, and activation of signaling pathways such as MYC, PI3K/AKT, WNT, NOTCH, HH, NF-κB, and others [410,411,412]. However, chemotherapy can also have unintended consequences. While it can kill cancer cells, dying cancer cells can release IL8, which activates self-renewal and regeneration of CSC, potentially leading to the recurrence of cancer cells, as observed in BC [413]. Although OC is initially sensitive to chemotherapy, a high recurrence rate is observed due to the development of chemoresistance, which is thought to be mediated by OCSC [82].

Therefore, there is a great need for appropriate treatment strategies that are effective against CSC and improve the outcome of cancer therapy. Several approaches have been proposed to target CSC, including the use of small molecules, antibodies, and immune-based therapies. Therapeutic approaches for targeting CSC heavily depend on the identification and isolation of CSC using cell-surface markers. However, targeting these markers poses a challenge due to their overlap with embryonic and adult stem cells, which can potentially affect normal adult stem cells and impair the process of normal tissue regeneration [414]. Furthermore, the shared signaling pathways and transcription factors between CSC and normal stem cells present an additional challenge [415]. Moreover, the scarcity of CSC within tumors adds to the difficulty of isolating and identifying them [180]. Another promising approach in cancer treatment is the use of combination therapies, in which drugs targeting different signaling pathways and cell types are used simultaneously to improve treatment efficacy. Some of the treatment strategies commonly used for BC and OC are listed in Table 2.

Specific biomarkers of BCSC and OCSC are being utilized in treatment strategies. Section 3 of the mentioned source suggests that anti-CD44 or anti-EpCAM antibodies could be employed for treating BCSC and OCSC. Notably, catumaxomab, an anti-EpCAM antibody, received approval from the European Commission in 2009 for the treatment of EpCAM-overexpressing tumors such as BC and OC [443,444]. The combination of catumaxomab and activated T cells shows promising potential as a powerful therapeutic approach for combating chemoresistant TNBC cells that express EpCAM [445]. Another approach involves the use of an anti-CD133 antibody, which has shown promise in OC and BC. A fusion protein comprising an anti-single-chain variable fragment (scFv) peptide sequence that targets the extracellular domain of human CD133 (anti-CD133scFV) and deimmunized PE38KDEL (dCD133KDEL) demonstrated reduced growth of the NIH:OVCAR5 OC cell line and inhibited tumor progression in xenografts [446]. In a recent review, Tume et al. explored the potential of anti-CD133 for BCSC treatment. While this treatment approach offers certain advantages, the authors also highlighted several challenges, suggesting that combining conventional treatments with anti-CD133 may be a promising strategy [447]. In a glioblastoma PDX model, the CD133-specific chimeric antigen receptor T cell (CAR-T) known as CART133 demonstrated remarkable anti-tumor efficacy. Notably, it did not exhibit any harmful effects on normal CD133+ hematopoietic stem cells in humanized CD34+ mice. These findings suggest that CART133 may hold promise for targeting other treatment-resistant CD133+ tumors [448].

A subset of OC and BC cases caused by mutations in *BRCA1* and *BRCA2* genes are sensitive to Poly (ADP-ribose) polymerase (PARP) inhibitors. Although PARP inhibition shows encouraging results even in platinum-resistant patients, resistance to PARP inhibitors can develop [449]. This resistance may be due in part to PARP inhibition inducing the CD133+ and CD117+ cell populations [450]. In BC, which are generally more resistant to PARP inhibitors, the combination of PARP inhibitor Niraparib and cyclin-dependent kinase inhibitor dinaciclib reduced EMT and cancer stem-like cell phenotypes [451]. The downregulation of RAD51, which is thought to mediate the resistance to PARP inhibitors, sensitized BCSC to PARP inhibition and resulted in reduced tumor growth [452]. 

ALM201, a 23-residue peptide derived from the N-terminal region of the FK506-binding protein like (FKBPL), is a promising candidate for targeting OCSC, as it specifically binds to the CD44 receptor and exhibits anti-angiogenic and anti-CSC activity. In several OC cell lines, ALM201 reduced tumorsphere-forming efficiency, but its efficacy was not consistent in vivo in a xenograft model due to differences in the microenvironment [427]. Another peptide derivative of FKBPL, AD-01, showed promising activity in BC, reducing ESA+/CD44+/CD24− and ALDH+ cell numbers, self-renewal ability, and expression of OCT4, NANOG, and SOX2 in BC cell lines and xenografts [453].

As the MEK1/2-ERK1/2 pathway is active in the majority of HGSOC cases, inhibition of this pathway was also tested as a potential therapeutic approach. MEK1/2 inhibitor trametinib showed an inhibitory effect on HGSOC proliferation and induction G0/G1 cell cycle arrest in these cells. However, it was also shown to promote stemness, as higher percentages of CD133+ cells were detected after trametinib treatment, as well as an increase in gene expression of stemness markers *SOX2*, *NANOG*, *OCT4*, and *ALDH1A* in vitro and in vivo. Therefore, the authors stress the possibility of treatment failure and/or resistance due to these mechanisms [454]. 

In a preclinical study, the PI3K inhibitor XL147 was used in combination with trastuzumab to treat trastuzumab-resistant BC. The results showed that the combination therapy reduced cancer cell proliferation and increased apoptosis. The treatment also decreased mammosphere formation, inhibited tumor growth, and reduced ALDH activity. This suggests that the addition of a PI3K inhibitor to trastuzumab treatment could potentially improve therapeutic outcomes for trastuzumab-resistant BC patients [455].

Repurposing of existing therapeutics has yielded some promising candidates for targeting CSC, including metformin, an antidiabetic drug; salinomycin, an antimicrobial agent commonly used in agriculture; and calcium channel blockers. Metformin has been shown to have the potential to inhibit both OCSC and BCSC, with low concentrations selectively targeting CD44+ CD117+ OCSC while not affecting OC cells, resulting in the inhibition of OCSC growth in vitro and in vivo [163,456]. In BC, treatment with metformin alone has been shown to decrease the CD44+/CD24− population of BCSC and reduce their ability to form mammospheres. When combined with doxorubicin, a synergistic effect was more effective in the reduction of tumor mass and prevention of relapse than either drug alone [3]. However, metformin-induced activation of AMPK has shown a context-dependent effect and a survival-promoting role in dormant ER^+^ breast cancer cells, the subpopulation of cancer cells responsible for therapy resistance and cancer recurrence [457].

Salinomycin has also been shown to be a selective inhibitor of BCSC and OCSC. It induces apoptosis in OCSC and selectively kills BCSC while reducing tumor growth and EMT [416,417,458]. However, its disadvantage is poor solubility in water and poor bioavailability, which can be remedied by conjugation with CD133-targeted nanoparticles [459]. Calcium channel blockers (manidipine, lacidipine, benidipine, and lomerizine) have been identified as good candidates because they inhibit AKT and ERK signaling, decrease stemness, and promote OCSC apoptosis [460]. Similarly, a calcium channel blocker (amlodipine) inhibits the growth of BC cell lines, leads to downregulation of p-ERK1/2, and inhibits colony formation [461].

Natural products are often explored as potential new therapeutics for cancer treatment. Shikonin (SHK), for example, induced apoptosis and inhibited migration, invasion, and xenograft tumor growth, as well as the expression of CSC-related markers (ALDH1, OCT4, SOX2, and NANOG) in OC [462]. A very similar effect was described for BC, where it was shown that the downregulation of CSC markers was modulated by the inhibition of STAT3, FAK, and SRC [463].

In recent years, the need for more precise and efficient therapy of CSC has increased—CSC vaccines for OC (targeting CD117+CD44+ OCSC) were able to activate immune responses to autologous tumor antigens, which in turn reduced OCSC, prolonged survival, and reduced tumor growth in mice [464]. This is consistent with evidence showing that CSC vaccines with high ROR1 expression effectively activate the immune response against OC [465,466]. Vaccines against BCSC are also being studied—vaccines containing human BCSC-lysates have been shown to prolong the lives of mice with BC. Other types of vaccines have also been found to target BCSC and reduce tumor growth (reviewed in [467]).

Overall, a better understanding of CSC biology and their interactions with the TME is needed to develop effective therapies that can improve cancer treatment outcomes.

## 9. Conclusions

Breast and ovarian cancers are among the most common and deadly cancers affecting women worldwide. BC and OC show marked intertumor heterogeneity, which subdivides them into various subtypes, and even intratumor heterogeneity. Although BC and OC arise from different tissues, they share approximately 50% of the most frequently mutated genes. Integrated molecular analysis revealed major differences between BC subtypes and many similarities between HGSOC and basal-like TNBC, suggesting a related etiology [38]. Even if the primary tumors initially respond well, recurrence is common in both BC and OC. Current research highlights the importance of CSC in tumorigenesis, as CSC play an important role in cancer initiation, progression, survival, metastasis, and chemoresistance. They can be distinguished by certain CSC-related markers. Herein, we highlighted the commonalities of BSCS and OCSC in terms of CSC marker expression, pathway activation, immune evasion, and microenvironment remodeling, and how this knowledge could be used for CSC-targeted therapy. Further advances in experimental models and the discovery of potential additional CSC-related markers or combinations thereof that are responsible for more aggressive phenotypes will enable the enrichment of CSC and a more targeted approach to study complex CSC biology. Therefore, to improve patient outcomes, further research is essential to unravel the complex role of CSC in BC and OC. Understanding the underlying mechanisms of CSC in BC and OC will be crucial for developing new treatment strategies.

## Figures and Tables

**Figure 1 ijms-24-10683-f001:**
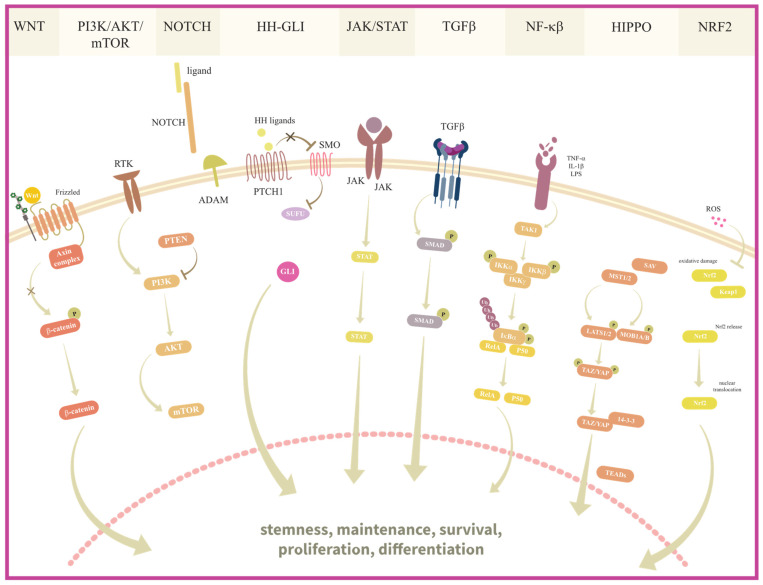
Common oncogenic signaling pathways that have been implicated in the maintenance of BCSC and OCSC. Picture shows canonical types of activation of nine oncogenic pathways: WNT, PI3K/AKT/mTOR, NOTCH, HH-GLI, JAK/STAT, TGFβ, NF-κβ, Hippo, and NRF2 signaling pathways.

**Figure 2 ijms-24-10683-f002:**
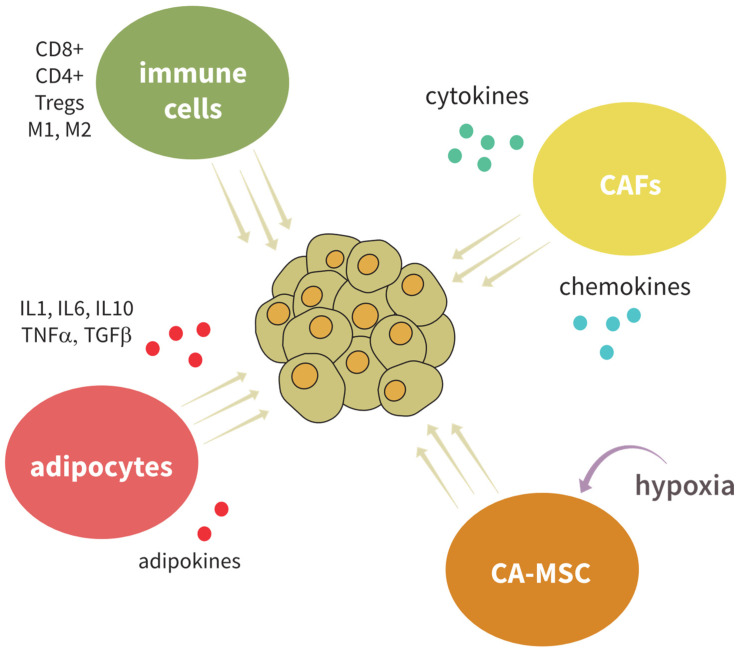
Tumor microenvironment of BC and OC. The most represented cell types are cancer-associated fibroblasts (CAF), adipocytes, immune cells, and carcinoma-associated mesenchymal stem cells (CA-MSC). All these cell types create a communication network and are accompanied by the secretion of different cytokines or chemokines which play an important role in determining the fate of the tumor.

**Table 2 ijms-24-10683-t002:** CSC-targeted strategies commonly used against BCSC and OCSC.

Compound	Mode of Action	Experimental Model	Effect
Salinomycin	Membrane ionophore antibiotic	BC cell lines	Reduces BCSC proportion, and improves survival [416].
Salinomycin-HDL		OC cell line (CD133+ population), normal ovarian epithelial line, xenografts	Downregulation of stemness markers (MYCN, NANOG, OCT4, SOX2) in vitro and in vivo, induction of apoptosis [417].
Metformin	Antidiabetic agent	BC cell lines (spheroids)	Decreases CD44+/CD24−/population and mammospheres formation [418].
		OC cell lines (spheroids), xenografts	Inhibits the CD44+ CD117+ population, no effect on CD44+ ALDH+ population, inhibits EMT [163].
Emodin	Anti-malarial and anti-allergic agent	BC cell lines and mouse models	Suppresses TGF-β1 production, reduces macrophage-induced EMT and BCSC formation, and reduces breast cancer lung metastasis [419].
		OC cell lines, xenografts	Inhibits the growth of OC cells in vitro and in vivo, reduces the number of CD44+/CD24− OCSC [420].
1α,25-dihydroxyvitamin D3	Vitamin	BC cell lines	Inhibition of spheroid formation, downregulation of stemness markers (OCT4, CD44, NOTCH1, NOTCH2, NOTCH3) [421].
		OC cell lines, xenografts	Inhibition of spheroid formation, downregulation of stemness markers (OCT4, CD44, NANOG, SOX2, KLF4, ABCG2), delays onset of tumor formation [422].
Luteolin	Flavonoid, antioxidant	BC cell lines, xenografts	Inhibition of stemness and chemoresistance via the NRF2-mediated pathway [423] and suppresses EMT and migration of TNBC cells by inhibition of Hippo/YAP pathway [424].
		OC cell line, primary OC (CD133+ ALDH+ populations), xenografts	Inhibition of stemness, inhibition of Hippo/YAP pathway, sensitization to paclitaxel and carboplatin [425].
ALM201	Anti-angiogenic therapeutic peptide	BC cell lines (spheroids), xenografts	Inhibition of spheroid formation, decrease in in vivo lung metastasis, and sensitization to tamoxifen by downregulating NOTCH4 and DLL4 [426].
		OC cell lines, xenografts	Inhibition of spheroid formation, inhibition of CD44+ CD117+ population, induction of differentiation [427].
Simvastatin	Inhibition of cholesterol synthesis	BC cell lines (spheroids), PDX	Radiosensitizes mammospheres of inflammatory BC (IBC) and TNBC, yet radioprotects HR+ and HER2+ [428]; inhibition of spheroid formation in PDX-derived TNBC [429].
		OC cell lines, primary cultures, xenografts	Inhibition of spheroid formation, downregulation of CD44 and ALDH1A1, reduction of tumorigenesis and metastasis in vivo [430].
All-trans retinoic acid (ATRA)	Vitamin A metabolite	BC cell lines, xenografts	Induces differentiation, inhibits invasiveness and migration of BCSC, and sensitizes to chemotherapy [431].
		OC cell lines (ALDH-high and ALDH-low), xenografts	Inhibition of stemness properties, inhibition of p62/NRF2 axis, decreases in vivo tumor growth only in ALDH-high OCSC [248].
Entinostat	HDAC inhibitors	BC cell lines, mouse model	Reduces CD44+/CD24− cell population, ALDH-1 activity, BMI-1, NANOG, and OCT4. Reduces tumor formation at the primary site and lung metastasis [432].
16cyc-HxA, 16lin-HxA, and 16KA		OC cell lines, xenografts	Induction of apoptosis in OCSC, reduction of tumor size in vivo [433].
Eugenol	Inhibition of NOTCH pathway	BC cell lines, mouse model	Downregulation of stemness of secondary mammosphere, lower CD44+/CD24−/low population, stemness suppression [434].
		OC cell lines, xenografts	Reduction of chemotherapy-induced spheroid formation, inhibition of CD44+ and ALDH+ cells, increases tumor-free survival [435].
Saracatinib + gemcitabine	Inhibition of SRC + chemotherapeutic	BC cell lines, mouse model	Synergistic effect, reverse drug resistance, inhibition tumor stemness, metastasis, and growth of BCSC (CD44+ OCT4+) [436].
Saracatinib + selumetinib	SRC inhibitor + MEK inhibitor	OC cell lines, primary cell lines, xenografts	Synergistic effect on cell cycle arrest, induction of apoptosis and autophagy, decrease in ALDH1+ population in vitro and in vivo [437].
Everolimus	Inhibitor of PI3K/Akt/mTOR signaling pathway	BC cell lines	Reverses Palbociclib resistance, decreases the expression of ALDH1 and NANOG, decreases cell migration, self-renewal, and EMT [438].
5-aza-2′-deoxycytidine	Inhibitor of DNA methylation	BC cell lines	Decreases the expression of CD44+/CD24−, activation of P53 expression, increases sensitivity to chemotherapy [439]
Everolimus + 5-aza-2-deoxycytidine	mTOR inhibitor + inhibitor of DNA methylation	OC cell lines, xenografts	Synergistic effect on OC tumorspheres (ALDH1+ CD44+) and tumor formation in vivo, induction of apoptosis [440].
Disulfiram + cisplatin + paclitaxel	ALDH inhibitor + chemotherapy	BC cell lines	Decreases ALDH activity, increases expression of SOX2 and NANOG. Disulfiram sensitizes BCSCs to chemotherapeutics [441].
		OC cell lines	Synergistic effect on OC cells, re-sensitization of cisplatin-resistant lines [442].

## Data Availability

Not applicable.

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
