# Peer review of "Deciphering Common Traits of Breast and Ovarian Cancer Stem Cells and Possible Therapeutic Approaches"

_ijms, 2023, doi:10.3390/ijms241310683_

Round 1
Reviewer 1 Report
In the review entitled “Deciphering common traits of Breast and ovarian cancer stem cells and possible therapeutic approaches”, the authors extensively highlight the shared features of BSCS and OCSC in terms of CSC marker expression, pathway activation, immune evasion, and the microenvironment remodeling. The manuscript is well-organized and includes relevant information about BC and OC common traits.
Although the manuscript provides a useful guide for BC and OC common features, the authors fail to address their main goal: provide possible therapeutic approaches based on their shared characteristics. And, in this sense, the authors should have inferred more about how the knowledge available for both OC and BC, can be used for CSC-targeted therapy. It seems more like a compilation of information lacking a critical point of view. The authors just collected the information and only in some sections, do they explain how this information is valuable in practical terms. An example of this compilation is the 2nd paragraph of section 8 (“8. CSC-targeted therapies”) which reports information that is completely out of the scope of this section. Please remove this paragraph from this section.
In the references section, there are some references missing details such as page numbers, volume numbers, or even DOI. Please review this section carefully and provide all the necessary information.
Minor corrections
In line 533, add T to the CD8+ cells (“Interaction between MCF7 cells and CD8+ T cells”).
In line 582, define CTL and Th1.
Between lines 643 and 644, there is a separation that should not exist (line 644 seems to be the beginning of a new sentence but is the remaining of the previous sentence).
Line 733, instead of “adipocyte gels” should be “adipocyte cells”, and the verb tense at the end of the sentence should be in the past and not in the present (“were” instead of “was”).
Author Response
We appreciate the Reviewer’s valuable input and are grateful for his/hers careful reading of our review. The comments have helped us enhance the manuscript and deliver a more comprehensive analysis of the subject matter. All revisions are marked in yellow.
In the review entitled “Deciphering common traits of Breast and ovarian cancer stem cells and possible therapeutic approaches”, the authors extensively highlight the shared features of BSCS and OCSC in terms of CSC marker expression, pathway activation, immune evasion, and the microenvironment remodeling. The manuscript is well-organized and includes relevant information about BC and OC common traits.
Response: We appreciate the Reviewer’s positive comments about our manuscript. We are glad to hear that the Reviewer found the manuscript well-organized and informative in terms of BC and OC common traits.
Although the manuscript provides a useful guide for BC and OC common features, the authors fail to address their main goal: provide possible therapeutic approaches based on their shared characteristics. And, in this sense, the authors should have inferred more about how the knowledge available for both OC and BC, can be used for CSC-targeted therapy. It seems more like a compilation of information lacking a critical point of view. The authors just collected the information and only in some sections, do they explain how this information is valuable in practical terms. An example of this compilation is the 2nd paragraph of section 8 (“8. CSC-targeted therapies”) which reports information that is completely out of the scope of this section. Please remove this paragraph from this section.
Response: We have carefully reviewed your comments and made revisions accordingly. We understand that our main goal should be to provide possible therapeutic approaches based on the shared characteristics of BCSCs and OCSCs. In the revised version, we have placed more emphasis on discussing how the knowledge available for both OC and BC can be translated into CSC-targeted therapies. We have provided further inferences and explanations on the practical implications of the information presented throughout the review.
Furthermore, we agree with the Reviewer’s comment regarding the second paragraph of section 8. It has been removed as suggested.
In the references section, there are some references missing details such as page numbers, volume numbers, or even DOI. Please review this section carefully and provide all the necessary information.
Response: We appreciate the Reviewer notifying us about the missing details in the references section, and we apologize for any inconvenience caused. It is possible that these mistakes occurred inadvertently while using various reference organizing programs. To address this issue, we have carefully revised all the references in Mendeley, using the style for the International Journal of Molecular Sciences (IJMS).
Minor corrections
In line 533, add T to the CD8+ cells (“Interaction between MCF7 cells and CD8+ T cells”).
Response: We have corrected it. Thank you.
In line 582, define CTL and Th1.
Response: We have defined CTL and Th1.
Between lines 643 and 644, there is a separation that should not exist (line 644 seems to be the beginning of a new sentence but is the remaining of the previous sentence)
Response: Thank you. We have corrected it.
Line 733, instead of “adipocyte gels” should be “adipocyte cells”, and the verb tense at the end of the sentence should be in the past and not in the present (“were” instead of “was”).
Response: Thank you. We have corrected it.
Thank you once again for your valuable feedback, and we hope that the revised version addresses your concerns and meets your expectations.
Reviewer 2 Report
The present article, written by Lučić and colleagues, provides an extensive review on recent progresses on the role of breast and ovarian cancer stem cells general understanding and on novel therapeutic approach aiming to target this resilient population. The topics are well addressed producing a coherent argument about the subject and a focused description of the field.
In my opinion the “4. Signaling pathways in BCSC and OCSC” needs a better introductory background regarding the described pathways as it has been done for the WNT signaling.
Several inhibitors of the above-described pathways affect CSC. A brief description should be added where possible.
The paragraph 5 is too long and should be divided in two or three subparagraphs.
Please spell out the full term at its first mention, indicate its abbreviation in parenthesis and use the abbreviation from then on and be consistent in their use (e.g. ovarian cancer).
A more insightful regarding the current limitations in treating CSC.
Author Response
We appreciate the Reviewer's input, which improved our manuscript significantly. All revisions are marked in yellow. Thank you for your valuable feedback.
The present article, written by Lučić and colleagues, provides an extensive review on recent progresses on the role of breast and ovarian cancer stem cells general understanding and on novel therapeutic approach aiming to target this resilient population. The topics are well addressed producing a coherent argument about the subject and a focused description of the field.
Response: We are grateful for the Reviewer's positive feedback on our article. It brings us great delight to learn that he/she found our review to be extensive and well-addressed, with a coherent argument and focused description of the role of breast and ovarian cancer stem cells, as well as novel therapeutic approaches targeting this resilient population. The Reviewer's recognition of our efforts in summarizing recent progress in the field and providing valuable insights is truly appreciated.
In my opinion the “4. Signaling pathways in BCSC and OCSC” needs a better introductory background regarding the described pathways as it has been done for the WNT signaling.
Response: We appreciate the Reviewer’s comment. We have added more introductory information about the described pathways.
Several inhibitors of the above-described pathways affect CSC. A brief description should be added where possible.
Response: As suggested, we have added additional information throughout the review. Thank you.
The paragraph 5 is too long and should be divided in two or three subparagraphs.
Response: We agree with the Reviewer. We have subdivided this paragraph into three subparagraphs.
Please spell out the full term at its first mention, indicate its abbreviation in parenthesis and use the abbreviation from then on and be consistent in their use (e.g. ovarian cancer).
Response: We have revised our manuscript as suggested.
A more insightful regarding the current limitations in treating CSC.
Response: We appreciate the Reviewer’s comment and have added more information about the challenges in treating CSC.
Thank you again for your valuable feedback. We appreciate it greatly. We hope the updated version addresses your concerns and meets your expectations.